# Integrated Transcriptome Analysis Reveals the Crucial mRNAs and miRNAs Related to Fecundity in the Hypothalamus of Yunshang Black Goats during the Luteal Phase

**DOI:** 10.3390/ani12233397

**Published:** 2022-12-02

**Authors:** Miaoceng Han, Chen Liang, Yufang Liu, Xiaoyun He, Mingxing Chu

**Affiliations:** 1Key Laboratory of Animal Genetics, Breeding and Reproduction of Ministry of Agriculture and Rural Affairs, Institute of Animal Science, Chinese Academy of Agricultural Sciences, Beijing 100193, China; 2College of Animal Science, Shanxi Agricultural University, Jinzhong 030801, China

**Keywords:** miRNAs-mRNAs, fecundity, goats, hypothalamus, luteal phase

## Abstract

**Simple Summary:**

The hypothalamus plays an important role in goat reproduction. In this research, RNA sequencing technology was used to detect and identify microRNAs (miRNAs) related to goat fertility in the goat hypothalamus. The results showed that several key miRNAs may affect the expression of their target genes in different ways with the high-fecundity goats during the luteal phase (LP-HF) vs. low-fecundity goats during the luteal phase (LP-LF), thereby affecting the key reproductive processes. These results reveal the mechanism on the reproductive performance of different litter sizes during the luteal phase of goats, from the perspective of the hypothalamus.

**Abstract:**

A normal estrus cycle is essential for the breeding of goats, and the luteal phase accounts for most of the estrus cycle. The corpus luteum (CL) formed during the luteal phase is a transient endocrine gland that is crucial for the reproductive cycle and pregnancy maintenance, and is controlled by many regulatory factors. However, the molecular mechanism of the hypothalamus effect on the reproductive performance of different litter sizes during the luteal phase of goats has not been elucidated. In this study, RNA-sequencing was used to analyze the mRNA and miRNA expression profiles of the hypothalamic tissues with the high-fecundity goats during the luteal phase (LP-HF) and low-fecundity goats during the luteal phase (LP-LF). The RNA-seq results found that there were 1963 differentially expressed genes (DEGs) (890 up-regulated and 1073 down-regulated). The miRNA-seq identified 57 differentially expressed miRNAs (DEMs), including 11 up-regulated and 46 down-regulated, of which 199 DEGs were predicted to be potential target genes of DEMs. Meanwhile, the functional enrichment analysis identified several mRNA-miRNA pairs involved in the regulation of the hypothalamic activity, such as the common target gene *MEA1* of novel-miR-972, novel-miR-125 and novel-miR-403, which can play a certain role as a related gene of the reproductive development in the hypothalamic–pituitary–gonadal (HPG) axis and its regulated network, by regulating the androgen secretion. While another target gene *ADIPOR2* of the novel-miR-403, is distributed in the hypothalamus and affects the reproductive system through a central role on the HPG axis and a peripheral role in the gonadal tissue. An annotation analysis of the DE miRNA-mRNA pairs identified targets related to biological processes, such as anion binding (GO:0043168) and small molecule binding (GO: 0036094). Subsequently, the KEGG(Kyoto Encyclopedia of Genes and Genomes) pathways were performed to analyze the miRNA-mRNA pairs with negatively correlated miRNAs. We found that the GnRH signaling pathway (ko04912), the estrogen signaling pathway (ko04915), the Fc gamma R-mediated phagocytosis (ko04666), and the IL-17 signaling pathway (ko04657), etc., were directly and indirectly associated with the reproductive process. These targeting interactions may be closely related to the reproductive performance of goats. The results of this study provide a reference for further research on the molecular regulation mechanism for the high fertility in goats.

## 1. Introduction

The traits of mammalian fecundity are determined by complex regulatory factors, including genetic material, nutritional level and feeding environment. The high performance of reproduction depends on the coordinated synthesis and release of hormones in the gonadal axis [1]. The hypothalamic–pituitary–gonadal (HPG) axis is a branch of the endocrine system, which controls the secretion of sex hormones and is the basis of the endocrine control for the reproduction in mammals [2]. The hypothalamus, located at the apex of the HPG axis, contains neurons that secrete a gonadotropin-releasing hormone (GnRH), which can produce GnRH signals and regulate the secretion of the downstream hormones, including the follicle-stimulating hormone (FSH) and the luteinizing hormone (LH) [3]. It can be said that reproduction is controlled by the hypothalamus, which is the key area in the brain to initiate reproductive activities. Alterations in the hypothalamus function, also affects the reproductive activity, including the follicular development, ovulation, and oviposition [4]. The corpus luteum is formed from a ruptured follicle during the luteal phase after ovulation [5]. The formation of the corpus luteum is crucial for the steroid biosynthesis and maintenance of early pregnancy [6], while sex steroid hormones, such as estrogen, progesterone, and androgen, play a key role in gender differentiation, reproductive function, and the sexual behavior of mammals [7].

Furthermore, reproduction is a complex process that is regulated by many genes, transcription factors, and regulatory non-coding RNAs. Since the Booroola fecundity gene (*FecB*) was identified in Booroola sheep [8], a series of studies have been carried out on the major genes controlling the sheep and goat follicle development, estrus, and ovulation by using molecular biology techniques. It has been found that deiodinase type 2 (*DIO2*) and deiodinase type 3 (*DIO3*) are involved in the regulation of the seasonal estrus in goats [9]. There were also other genes, retinol-binding protein 4 (*RBP4*), GnRH and its receptor *GnRHR*, which are related to the reproduction in goats. In addition, kisspeptin is a polypeptide encoded by the *Kiss-1* gene, which is highly expressed in two major neuronal populations of the hypothalamus. It is a recognized reproductive hormone coordinator, and the kisspeptin/GnRH pathway is very important for the reproductive endocrine system [10]. Moreover, different molecules that play an epigenetic role, such as miRNA, were also investigated in this context. MicroRNAs (miRNAs) can bind to complementary mRNAs and influence the post-transcriptional level of genes by targeting or inhibiting the expression of the transcripts [11]. There are many miRNAs acting as mediators to further regulate the activity of the HPG axis [12]. So far, miR-7a, let-7a, and miR-30b had been found to regulate the reproductive activity. The miR-7a can be co-expressed with agouti-related peptide (*AgRP*) in AgRP neurons [13] and plays an important role in reproduction by acting on the mTOR pathway within the AgRP neurons [14]. One study reported that the deletion of miR-7a led to hypogonadotropic hypogonadism in mouse models [15]. Let-7a and its related protein lin-28b are key components of the neuroendocrine mechanisms that control the timing of the onset of puberty, regulating reproduction by acting on kisspeptin and/or GnRH neurons [16,17]. The miR-30b is expressed in kisspeptin ARC(arcuate nucleus) neurons and regulates the kisspeptin expression by binding to the makorin ring finger protein 3 (*MKRN3*) gene, thereby inhibiting its activity [18].

In the development of human civilization, goats played important roles in human agricultural and economic development by providing meat, milk, skin, hair, and other products. The reproductive trait is one of the main economic traits of the goat, and the reproductive performance is directly related to the economic benefits of the goat industry. Therefore, it is important to elucidate the interaction mechanism of mRNAs and miRNAs during goat reproduction, to improve the reproduction performance of goats. However, the molecular mechanisms involved in the regulation of the high-fecundity traits in the hypothalamus tissue during the luteal phase of goats, are still poorly understood. The Yunshang black goat is a new black goat breed bred for meat, with a high fertility in China, which has the characteristics of a fast growth, good meat production performance, and excellent meat quality. In this study, the high-fecundity goat breed was used as the research object, and the RNA-seq technology was used to analyze the expression profiles of miRNAs and mRNAs in the hypothalamus during the luteal phase of high- and low-fecundity goats. The key molecular mechanisms and molecular networks of the breed’s fecundity were revealed. The purpose of this study was to help our understanding of the molecular mechanism of the prolific traits. The results of this study provided a useful resource for further research on miRNAs and mRNAs in high-fecundity goats and their potential relationships and functional roles in the reproductive regulation.

## 2. Materials and Methods

### 2.1. Ethics Statement

The experimental animals in this study were in compliance with animal welfare standards. The animal study under a permit (No. IAS2019-63) was reviewed and approved by the Institutional Animal Care and Use Ethics Committee of the Institute of Animal Sciences and Chinese Academy of Agricultural Sciences (IAS-CAAS) (Beijing, China) for all of the experimental procedures mentioned. All protocols complied with animal welfare guidelines and minimized animal suffering.

### 2.2. Goat Selection and Sample Collection

Ten Yunshang black nanny goats, aged 3–5 years (The average age of each low-fecundity and high-fecundity group was 4 years), with no significant difference in weight, height, and physical condition, were selected as the research objects from the goat core breeding farm in Kunming, Yunnan Province. In addition, all goats were raised under the same conditions with free access to water and feed. All of the experimental goats were treated for synchronous estrus, and a CIDR (controlled internal drug-release device) suppository (30 mg flurogestone acetate) was placed in the vagina of the goats for 16 days and then removed. On the 11th day after the CIDR removal, it was the luteal phase, and slaughter and hypothalamus sampling were performed. The specific process is shown in Figure 1. The goats were divided into high fecundity goats at luteal phase (LP-HF) group (n = 5, average litter size: 3.40 ± 0.42) and low fecundity goats at luteal phase (LP-LF) group (n = 5, average litter size: 1.80 ± 0.27) according to their kidding number. The collected hypothalamic tissues were washed with normal saline, placed in the cryopreservation tubes, and immediately placed in liquid nitrogen for short-term storage, and then brought back to the laboratory and placed in a −80 °C refrigerator for long-term storage and the subsequent experiments.

### 2.3. Total RNA Extraction

According to the manufacturer’s instructions, the total RNA was extracted from the hypothalamus tissue samples of 10 Yunshang black goats with TRIzol reagent (Thermo Fisher Scientific, Waltham, MA, USA) and used for the RNA sequencing [19]. High-quality RNA is the basis for the successful sequencing. We used the following methods to detect the samples, and the library can be constructed only after the test results meet the requirements:

(1) The Nanodrop 2000 spectrophotometer (Thermo Scientific, Wilmington, DE, USA) was used to detect the purity (OD 260/280), concentration, and nucleic acid absorption peak of RNA;

(2) The Agilent 2100 RNA Nano 6000 Assay Kit (Agilent Technologies, Palo Alto, CA, USA) was used to accurately detect the integrity of RNA;

(3) We used 1% agarose gel electrophoresis to detect whether the RNA sample was degraded or contaminated with genomic DNA.

### 2.4. Library Preparation and Sequencing

To begin, 3 μg of total RNA was taken from each hypothalamic sample for the mRNA library construction. The library construction was performed, according to the instructions for the NEB Next^®^ Ultra TM Directional RNA Library Prep Kit and Illumina^®^ (NEB, Ipswich, MA, USA). Then, the library was sequenced on the Illumina Novaseq6000 platform, to generate 150 bp paired-end (PE150) sequence reads. The sequence reads in fastq format were first filtered by the SOAPnuke (v2.1.0) [20] and HISAT2 (v2.1.0) [21] was used to map the clean reads to the reference genome (GCF_001704415.1). Subsequently, we used String Tie (v1.3.5) [22] to assemble the transcripts, and performed the fragments per kilobase per base (FPKM) transformation to obtain the expression levels of the transcript. 

Similarly, 3 μg of total RNA was used for miRNA library construction, the library was sequenced on the Illumina Hiseq 2500 platform, resulting in 50 bp single-end (SE50) reads. Using the clean reads of 18–35 nt in length for the subsequent analyses, and small RNA (sRNA) was aligned to the reference genome (GCF_001704415.1) by bowtie (v1.0.1) [23] to identify the known miRNAs. In this process, the miRbase (v22.0) database [24] was used as a reference. Referring to mirEvo (v2.0.0.5) and miRdeep (v2.0.0.5) [25], novel miRNAs were predicted, based on the signature hairpin structures of the miRNA precursors. The expression amount of the known and novel miRNAs in each sample was counted. The expression levels were normalized with TPM (Transcripts per million).

### 2.5. Analysis of the Differential Expression mRNAs and miRNAs

We used DESeq2 (v3.18.1) [26] to analyze the differentially expressed mRNAs in LP-HF and LP-LF. The default settings for the screening threshold are |log_2_FC| > 1 and *q*-value < 0.05. Meanwhile, based on the TPM expression levels of miRNAs, we also used DESeq2 to identify the DEMs of LP-HF and LP-LF, with |log_2_FC| > 1, *p*-value < 0.05 as the differential expression. 

We input the differential gene expression values of the high- and low-fecundity comparison groups, and use the log10 as the base to process the data. We used the pheatmap package of the R software (v 4.2.1) (https://www.r-project.org, accessed on 1 May 2022) to read the data for drawing the heat map, in order to visually present the global expression changes and clustering relationships of multiple genes in multiple samples.

### 2.6. Gene Ontology and Kyoto Encyclopedia of Genes and Genomes Analyses

The functional annotation and pathway enrichment analysis of the DEGs and the predicted target genes of the DEMs were performed using GO [27] (Gene Ontology, http://www.geneontology.org (accessed on 3 November 2022), of GOseq (v2.12) software and KEGG [28] (Kyoto Encyclopedia of Genes and Genomes) of KOBAS (v2.0) software, to visualize the data. The GO analysis mainly includes a cellular component (CC), molecular function (MF) and biological process (BP). The GO items and KEGG pathways significantly enriched by the DEGs, were calculated using hypergeometric distribution, and the GO items and KEGG pathways with a *q*-value < 0.05 (If *q*-value < 0.05 was used to enrich the very few GO items and KEGG pathways, then *p*-value < 0.05 was used for the enrichment) were considered significantly enriched.

### 2.7. Integrated miRNA-mRNA Co-Expression Network Analysis

The MiRanda v3.3a and qTar (https://github.com/zhuqianhua/qTar.git, accessed on 28 April 2020) were used to predict the target genes of the miRNAs. Then, we analyzed the interactions between the miRNAs and mRNAs. Based on the miRNAs function, the mRNAs that were negatively related to the miRNAs were screened out, in order to accurately identify the key association with the reproductive DEMs and DEGs, and the miRNA-mRNA interaction networks were built by using Cytoscape (v3.8.2, http://www.cytoscape.org/, accessed on 1 May 2016).

### 2.8. RT-qPCR Validation

In order to validate the accuracy of the sequencing data, six mRNAs, including *NCAM1*, *FGFR2*, *LINGO1*, *MEA1*, *SDCCAG8,* and *MX2*, and six miRNAs, including chi-miR-200a, chi-let-7d-3p, chi-miR-10b-3p, chi-let-7b-3p, novel-miR-403, and novel-miR-1125, were randomly selected for the data validation. The primers (Table 1) of the mRNAs and miRNAs were synthesized by Beijing Tianyi Huiyuan Biotechnology Co., Ltd. (Beijing, China) for the subsequent reverse transcription. *RPL19* and *U6* were used as reference genes for mRNA and miRNA, respectively.

For the RT-qPCR analysis of the mRNAs, the RT-qPCR with the SYBR Green qPCR Mix (TaKaRa, Dalian, China) was conducted with a Roche Light Cycler^®^ 480 II system (Roche Applied Science, Mannheim, Germany). For the miRNAs, the RT-qPCR was conducted by miRcute Plus miRNA qPCR Kit (TIANGEN, Beijing, China).

For the specific operation steps, please refer to the supplier’s manual. In addition, *RPL19* (for mRNA) and *U6* small nuclear RNA (for miRNA) were utilized as the reference gene-miRNA to calculate the relative expression level with the method of 2^−ΔΔCt^.

### 2.9. Statistical Analysis

We performed the statistical analysis of the RT-qPCR results and graphs using GraphPad Prism (v 5.0) software. The statistical data were tested by performing paired *t*-tests. The results are presented as the means of three replicates, then the gene expression levels were determined and compared with the RNA-seq data.

## 3. Results

### 3.1. The cDNA Library Sequencing and mRNA Transcriptome Analysis

Ten cDNA libraries were sequenced by Illumina, and the RNA-seq for the mRNA generated approximately 1111 million clean reads after the data filtering (Table 2). More than 96.15% of the clean reads were mapped to the goat reference genome. The GC(guanine-cytosine content) content ranged from 41.3 to 49.0%, and the scores of Q20 and Q30 were above 97.5% and 92.8%, respectively. This indicated that the reliability and quality of the sequencing data were sufficient for further analysis. A total of 72,220 mRNAs were identified after mapping the goat genome (Appendix A). Regarding the expression levels of the mRNAs, our results showed that genes with a high RNA-seq expression (genes with FPKM > 60) accounted for about 1.19% (Figure 2A), and the gene expression levels (FPKM) of the 10 hypothalamus tissues were similar (Figure 2B). 

### 3.2. Small RNA Library Sequencing and miRNA Transcriptome Analysis

A total of 249.7 million raw reads were obtained by the miRNA transcriptome sequencing in the hypothalamus from the luteal phase of the high and low fecundity Yunshang black goats, and 215.5 million clean reads remained after filtering. The clean reads were compared with the goat reference genome (Table 3), the alignment rate was about 97.52%, the length of the filtered sequences was between 18–35 nt (Figure 3A), the GC content was between 47.71% and 51.79%, and the average Q20 content was 99.22% and Q30 content was 97.38%, indicating that the quality of the data generated by sequencing was relatively high. In addition, a variety of non-coding RNAs were also identified (Appendix A), and most of the small RNAs were known miRNAs, accounting for about 73.29% (Figure 3B). Finally, a total of 1837 miRNAs were identified in our transcriptome data, including 424 known miRNAs and 1413 novel miRNAs for the subsequent analysis (Appendix A). Among them, the most abundant miRNAs in the hypothalamus from the luteal phase of Yunshang black goat include miR-9, let-7, and miRNA-200 family members.

### 3.3. Differential Expression and Functional Enrichment Analysis of the mRNAs in the LP-HF vs. LP-LF

In this study, the differential expression analysis of the mRNA of the high- and low-fecundity Yunshang black goats was carried out, with |log_2_FC| > 1 and *q* value < 0.05 as the screening criteria. In the LP-HF vs. LP-LF, a total of 1963 DEGs were screened. Among them, 890 DEGs were up-regulated and 1073 DEGs were down-regulated (Appendix A, Figure 4A). The heatmap of the DEGs in the LP-HF vs. the LP-LF comparison group showed that the DEGs had different expression patterns between the LP-HF and LP-LF (Figure 4C).

We analyzed the different functional categories of the GO and the enrichment of the KEGG pathway of 1963 DEGs in the hypothalamus. The most abundant terms in the LP-HF vs. LP-LF were the anion binding (GO:0043168) and the small molecule binding (GO: 0036094) (Appendix A, Figure 5A). The KEGG analysis showed that the DEGs were significantly enriched in 11 KEGG pathways, of which the hedgehog signaling pathway (ko04340) was the most enriched (Appendix A, Figure 6A). Moreover, we also discovered several pathways involved in the reproductive biology, such as the GnRH signaling pathway (ko04912), steroid hormone biosynthesis (ko00140) and estrogen signaling pathway (ko04915). Of the DEGs, we found several key genes involved in the reproduction process, such as cytochrome P450 family 19 subfamily a member 1 (*CYP19A1*), neural cell adhesion molecule 1 (*NCAM1*), and the fibroblast growth factor receptor (*FGFRs*) family.

### 3.4. Differential Expression and Functional Enrichment Analysis of the miRNAs in the LP-HF vs. LP-LF

A total of 57 DEMs were screened in the LP-HF vs. LP-LF groups. There were 11 up-regulated DEM expressions and 46 down-regulated DEM expressions (Appendix A, Figure 4B). Furthermore, a heatmap of the DEMs in the LP-HF vs. the LP-LF comparison group revealed that the DEMs had different expression patterns between the LP-HF and LP-LF (Figure 4D). In order to better understand the biological functions of the 57 identified DEMs, we predicted the potential target genes of these miRNAs (Appendix A), of which 14 known and 43 novel DEMs targeted 1953 and 12,936 genes, respectively.

The GO function enrichment analysis of the target genes of the DEMs was conducted (Appendix A, Figure 5B). In the LP-HF vs. the LP-LF comparison group, the most significantly enriched GO items are the synaptic vesicle membrane (GO: 0030672) and the exocytic vesicle membrane (GO: 0099501). In addition, the positive regulation of the synaptic vesicle transport (GO: 1902805) and the positive regulation of the synaptic vesicle recycling (GO: 1903423) were also enriched. The KEGG analysis (Appendix A, Figure 6B) showed that in the LP-HF vs. the LP-LF comparison group, the most significantly enriched pathways were the Apoptosis (ko04210) and the Notch signaling pathway (ko04330). In addition, the Jak-STAT signaling pathway (ko04630), the Cytokine-cytokine receptor interaction (ko04060) and the VEGF signaling pathway (ko04370) also had been enriched. Among them, 30 KEGG pathways were identified as significantly associated with the DEM target genes.

### 3.5. miRNA-mRNA Co-Expression Network Analysis

To further analyze the relationship between the miRNAs and mRNAs, we constructed an interacting co-expression network using the potential target genes of the DEMs and DEGs obtained by the RNA-Seq. The potential target genes of the 57 identified DEMs were intersected with 1963 DEGs, obtained from the RNA-seq, resulting in 199 intersecting genes (Appendix A, Figure 7A). Cytoscape was then used to construct an interaction network of the miRNA-mRNA pairs (Appendix A, Figure 7B).

The GO enrichment analysis was performed to reveal the biological process terminology of the enrichment (*q*-value < 0.05), to gain insight into the function of the miRNA-mRNA pairs with the negatively correlated expression between the LP-HF and LP-LF. The GO terms were shown in Table 4, and the most abundant terms in the LP-HF vs. LP-LF were the anion binding (GO:0043168) and the small molecule binding (GO: 0036094). Among them, the *MX2* gene was annotated into almost all GO terms, indicating that this gene has an important regulatory role in the process of this study, and further in-depth research is needed. In addition, we also performed the KEGG pathway analysis of the miRNA-mRNA pairs with negatively correlated miRNAs. The enriched KEGG pathways were shown in Table 5. The two top pathways were the Hedgehog signaling pathway (ko04340) and Ubiquitin mediated proteolysis (ko04120). Interestingly, the GnRH signaling pathway (ko04912), the estrogen signaling pathway (ko04915), Fc gamma R-mediated phagocytosis (ko04666), and the IL-17 signaling pathway (ko04657), were directly and indirectly associated with the reproductive process.

### 3.6. Data Validation

To evaluate the accuracy of the sequencing, we randomly selected the mRNAs and miRNAs from the sequencing data for the RT-qPCR validation. We measured gene expression levels and compared them with the RNA-seq data. The results showed that the RNA-seq data and the RT-qPCR data exhibited similar patterns (consistent up- and down-regulation relationships) (Figure 8), which indicated that the data generated from the RNA-seq were reliable. 

## 4. Discussion

The hypothalamus is a master regulator of reproduction. The GnRH neurons distributed in the hypothalamic tissue can secrete GnRH, and then GnRH can regulate the secretion of the gonadotropins, such as FSH and LH, by the pituitary gland, causing the secretion of related hormones and affecting the reproductive activities [3]. Strict regulation of the central nervous system and the endocrine system is necessary to achieve reproduction [29]. In this study, the whole transcriptome of the hypothalamus from the luteal phase of Yunshang black goats was established. The expression profiles of the miRNA and mRNA in the hypothalamus were revealed by comparing the reproductive performance of the high and low fecundities, and the miRNA-mRNA network was constructed. The miRNA-mRNA interaction network comprehensively investigated the molecular basis of the candidate genes involved in the goat reproductive traits. A total of 72,220 transcripts and 1837 miRNAs were identified in hypothalamic samples of the goats, including 1963 DEGs and 57 DEMs.

The miRNAs are the most widely distributed regulatory factors in animals. Studies have found that many miRNAs are involved in a variety of biological regulation processes in the hypothalamus, such as immune response [30], osmotic regulation [31], and so on. In this study, miR-9 is the most abundant miRNA in the hypothalamus of Yunshang black goats, during the luteal phase, which was consistent with the results of the follicular phase in the previous [32]. It has been reported that miR-9 is mainly expressed in neural tissues in mammals, such as mice and humans [33], and regulates *BAF53a* (also known as *ACTL6A*, actin like 6A) [34], stathmin [35] and other genes involved in the proliferation, migration, and other activities of the nerve cells. Some studies have found that in the hypothalamus of rats, the changes of lin28/let-7 may be involved in the initiation of puberty [17]. In mice, the expression pattern of miRNA-200 was associated with the increase of the GnRH expression in the hypothalamus before sexual maturity [36]. In addition, let-7 and miRNA-200 family members have been found to be highly conserved across species, in sequence and function, and also have a period specificity and fecundity specificity [37], which plays an important role in a variety of cell activities, including the cell cycle regulation and apoptosis [38]. In this study, let-7 and miRNA-200 family members were highly expressed in the hypothalamus of both high-fecundity and low-fecundity groups of Yunshang black goats, during the luteal phase.

In addition, 57 DEMs in the high and low fecundity groups of Yunshang black goats were identified. One of the highly conserved miR-10 family has been shown to be involved in the cell proliferation and apoptosis. Yu et al. found that members of the miR-10 family were highly expressed in goose follicles, during the laying and nesting stages [39]. Tu et al. found that both miR-10a and miR-10b can inhibit the expression of key factors in the brain-derived neurotrophic factor (*BDNF*) and the TGF-β signaling pathways, thereby inhibiting the follicular granule cell proliferation and induction of granulosa cell apoptosis [40]. Wei et al. found that miR-10b could target *BDNF* and inhibit the activity of goat ovarian granulosa cells by inhibiting the expression of *BDNF* [41]. However, due to the limited research on the functions of miRNAs in the hypothalamus, the deep functions of the miRNAs screened in this study need to be further studied.

In the functional analysis of the hypothalamic DEGs in the LP-HF vs LP-LF, we found that *CYP19A1*, *NCAM1*, and the *FGFRs* family were involved in the reproduction process. 

Aromatase, encoded by the *CYP19A1* gene, is an estrogen-synthesizing enzyme that catalyzed many reactions related to steroidogenesis and the conversion of androgens to estrogens. A study has shown that *CYP19A1* is involved in the follicle development and follicular atresia [42], and is also important in the sheep gonad development, horse testicular cells, and luteal development [43]. Sun et al. showed that *CYP19A1* was expressed in the ovary of Guangxi Ma chicken, during the laying period, and can promote the physiological function of the ovary and negatively regulate the hypothalamus during the process of estrus or ovulation [44]. A previous study on the *CYP19A1* gene had also focused on cancer-related diseases in humans [45]. This study selected Yunshang black goat individuals with different reproductive performances during the luteal phase, and found that *CYP19A1* was expressed in the hypothalamus. In addition, the expression level in the hypothalamus of goats with a high fecundity was higher than those with a low fecundity (*p* < 0.05), and the KEGG pathway enrichment analysis showed that the gene was enriched in the steroid hormone biosynthesis (ko00140) and the ovarian steroidogenesis (ko04913), that the pathways were associated with the reproductive function of the hypothalamus, and that was an important candidate gene.

*NCAM1* belongs to the immunoglobulin superfamily of the cell adhesion molecules and is widely present in the nerve cells of the central and peripheral nervous systems [46]. *NCAM1* is highly expressed in the hypothalamus, which is similar to the results of this study, especially in the glial cells of the GnRH nerve endings [47]. Furthermore, Rubinek et al. demonstrated that the homologous binding of *NCAM1* can induce the GH secretion in fetal pituitary cultures [48]. In addition, studies have shown that *NCAM1* can mediate the anti-apoptotic, pro-neurogenesis and neuroprotective biological effects of FGFs and GDNF by binding to *FGFR,* and the glial cell derived neurotrophic factor (*GDNF*) receptor *GFRα* [49,50]. The signal pathway enriched by KEGG was the cell adhesion molecules (ko04514). A study has shown that the intercellular communication mediated by the cell adhesion molecule signaling pathway plays an important role in regulating the expression and secretion of hormones, such as *GH* [51].

The family of *FGFRs* is composed of four tyrosine kinase receptors (*FGFR1*-*FGFR4*) that bind to the fibroblast growth factors (*FGFs*) to mediate the cell signaling. The binding of *FGFs*/*FGFR* can activate the complex downstream signals, such as PLCγ, MAPK, and PI3K-AKT, and play an important role in the process of the cell proliferation, differentiation, survival and migration [52], and specifically involved in promoting angiogenesis [53], injury repair [54], embryonic development [55], and nerve regeneration [56]. The binding of *FGF2* secreted by astrocytes to *FGFR1* can promote the differentiation and survival of GnRH neurons, and enhance the synthesis of the GnRH protein [57], thereby participating in the regulation of animal reproduction. *FGFR2* is a high-affinity receptor for most *FGFs*. Studies on animals and humans have shown that *FGFR2* is expressed on trophoblast membranes [58]. The inhibition of the *FGFR2* expression resulted in the reduced trophoblast formation and delayed the trophoblast growth [59]. Thus, the role of *FGFR2* in the placental development and fetal growth is important. Cells that express both *FGFR1* and *FGFR2* can receive more *FGF4* signals and exhibit a higher ERK activity [60]. During development, the FGF-ERK signaling pathway plays an important role in the proliferation, differentiation, and apoptosis.

Overall, the genes mentioned above play important roles in the regulation of animal reproduction. The KEGG enrichment analysis found that these DEGs were significantly enriched in the signal transduction, nervous system, endocrine system, metabolism, and other related pathways. These pathways play crucial roles in the biosynthesis of reproductive hormones. We found that many GO terms related to reproductive biology were also enriched, such as the steroid hormone mediated signaling pathway (GO:0043401), the response to the steroid hormone (GO:0048545) and the cellular response to the steroid hormone stimulus (GO:0071383).

The typical molecular mechanism of the miRNA is to exert its regulatory effect through target genes. An integrative analysis of mRNAs and miRNAs by the RNA-Seq was used to construct regulatory networks to further understand the key genes affecting the reproductive processes.

Among the 57 miRNAs and predicted target genes differentially expressed during the luteal phase, the common target gene of the novel-miR-972, novel-miR-125, and novel-miR-403, was the male-enhanced antigen 1 (*MEA1*). A study has shown that *MEA1* plays a certain role as a gene related to the reproductive development in the HPG axis and its regulatory network, by regulating the androgen secretion [61]. The target gene *SDCCAG8* of the novel-miR-972 was found in a previous study at our laboratory [62], indicating that *SDCCAG8* was a high-fecundity convergent gene in sheep and goats, and can affect the litter size of goats through the TGFβ pathway. Another target gene predicted by the novel-miR-403 was adiponectin receptor 2 (*ADIPOR2*). A study has shown that *ADIPOR2* is widely distributed in many reproductive organs, including the central nervous system, ovaries, and fallopian tubes [63]. There is evidence that adiponectin affects the reproductive system through the central effects on the HPG axis, as well as the peripheral effects on the gonadal tissue [64]. In addition, adiponectin also affects the secretion of oxytocin, which has a profound effect on the HPG axis and the reproductive function [65]. The novel-miR-1125 targets *LINGO-1*, a protein encoded by a leucine-rich repeat (LRR), neurite outgrowth inhibitory protein (Nogo), and immunoglobulin domain (Ig) constitutes the central nervous system transmembrane protein. In a variety of animal models, the targeted inhibition of *LINGO-1* can promote the neuronal survival, axon regeneration, and the oligodendrocyte differentiation [66].

Meanwhile, miR-1271 was involved in important biological functions, such as neuro-regulation. The overexpression of miR-1271 can inhibit the cell proliferation and induce apoptosis [67]. In this study, the miRNA was highly expressed in the hypothalamus, speculating that it could regulate the reproductive performance of goats by affecting the proliferation of nerve cells. A previous study of our subject also demonstrated that miR-1271 targeting *TXLNA* in ovarian tissue, was a potential regulatory pathway affecting the reproductive performance of goats [68]. The interaction network in this study showed that miR-1271 targets *PNN* interacting serine and the arginine rich protein (*PNISR*). *PNISR* is a part of the multi-protein complex in the nucleus and is involved in the processing of pre-mRNA to affect the cell proliferation and differentiation [69]. In addition, another target gene was MX Dynamin like GTPase 2 (*MX2*), which has been shown to significantly inhibit the cell proliferation by prolongating the cell cycle [70]. MX2 protein has also been reported as a biomarker for the early gestation diagnosis in buffalos [71]. In conclusion, miR-1271 is worthy of further investigating.

The predicted target genes of the DEMs in high-fecundity and low-fecundity Yunshang black goats, during the luteal phase, were in Apoptosis (ko04210), the Notch signaling pathway (ko04330), Fc gamma R-mediated phagocytosis (ko04666), the Cytokine-cytokine receptor interaction (ko04060), the Jak-STAT signaling pathway (ko04630), cell adhesion molecules (ko04514), and other 30 KEGG pathways, which were significantly enriched. At the same time, both the differential genes and miRNA predicted target genes were enriched in two signaling pathways, the mTOR signaling pathway (ko04150) and the ErbB signaling pathway (ko04012). Studies have shown that the pathways and the sexual maturation initiation are closely related [72,73].

## 5. Conclusions

In conclusion, we analyzed the miRNA and transcriptome profiles in the hypothalamus of Yunshang black goats, during the luteal phase, to identify the key genes and miRNAs that may be involved in the reproductive process, and constructed the complete mRNA-miRNA interaction of the Yunshang black goat. We identified several DEGs (such as *CYP19A1*, *NCAM1,* and *FGFRs*) and miRNA-mRNA pairs (including *MEA1* that was co-expressed with the novel-miR-972, novel-miR-125, and novel-miR-403). These miRNAs and genes may play important roles in the regulation of the goat hypothalamus, during the luteal phase, which is worthy of further study in the reproductive regulation of goats and the multiple birth mechanism, in the future. Further studies should validate the functions of these key genes and the interaction networks at the cellular level of the goat fecundity.

## Figures and Tables

**Figure 1 animals-12-03397-f001:**
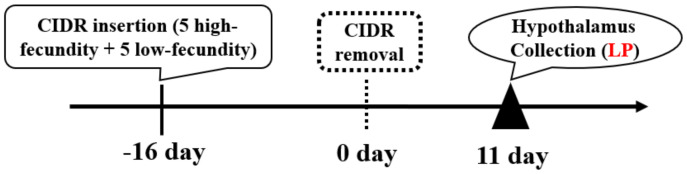
Simultaneous estrus processing and sampling. LP denotes the luteal phase.

**Figure 2 animals-12-03397-f002:**
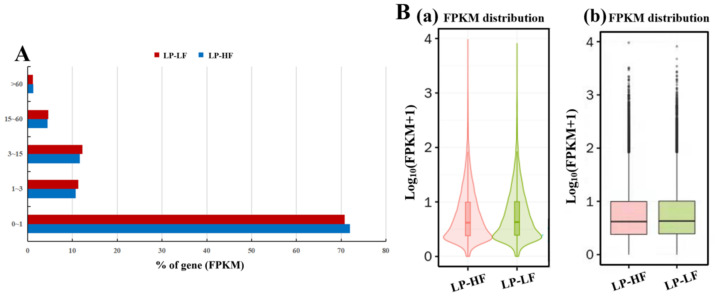
FPKM (**A**) distribution of the identified genes and violin (**a**), box plots (**b**) of the distribution of the FPKM expression levels (**B**) across all transcripts under the LP-HF and LP-LF comparisons. The width of each violin graph reflects the number of transcripts at that expression level.

**Figure 3 animals-12-03397-f003:**
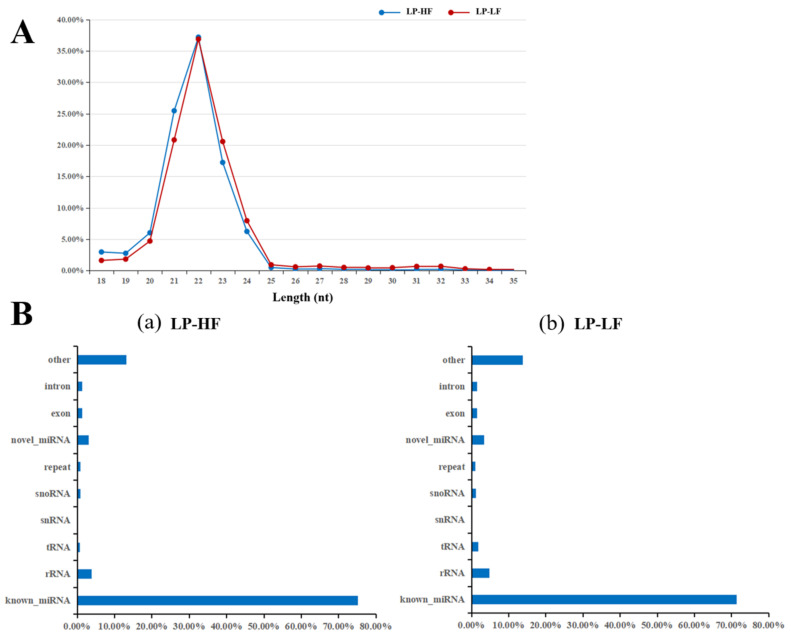
Length distribution of the small RNA reads and the percentage of the detected miRNAs from the non-coding RNAs (ncRNAs). (**A**) Length distribution of the clean reads from the sRNA fragments. (**B**) Categories of identified ncRNAs by sequencing in the LP-HF (**a**) and LP-LF (**b**).

**Figure 4 animals-12-03397-f004:**
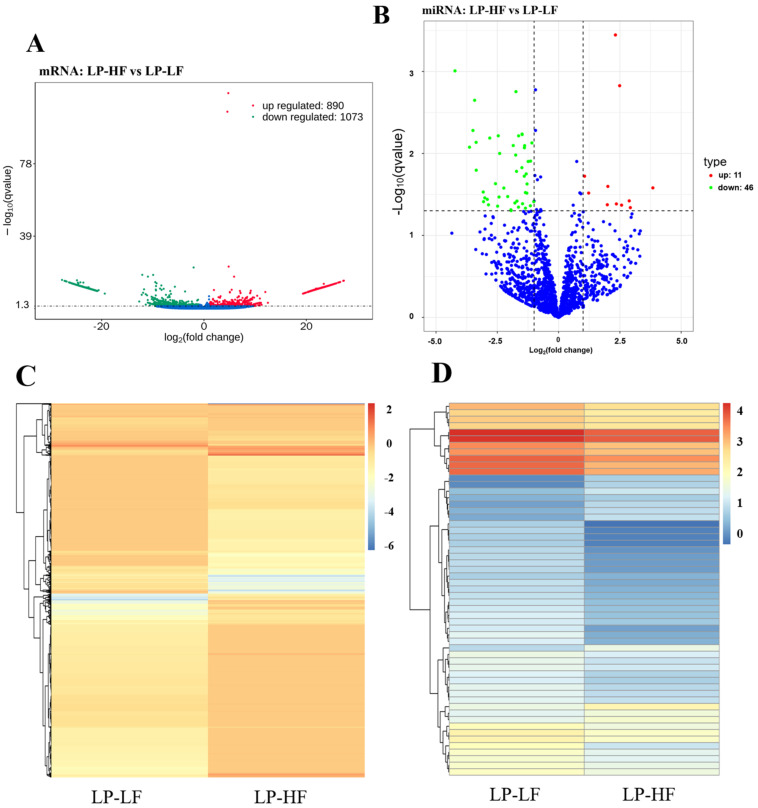
DEGs and DEMs analysis. Volcano plot of the identified genes (**A**) and the miRNAs (**B**) in the LP-HF vs LP-LF, where red and green represent the up- or down-regulation, respectively. Heat maps showing the expression patterns of 1963 DEGs (**C**) and 57 DEMs (**D**) in the LP-HF vs LP-LF.

**Figure 5 animals-12-03397-f005:**
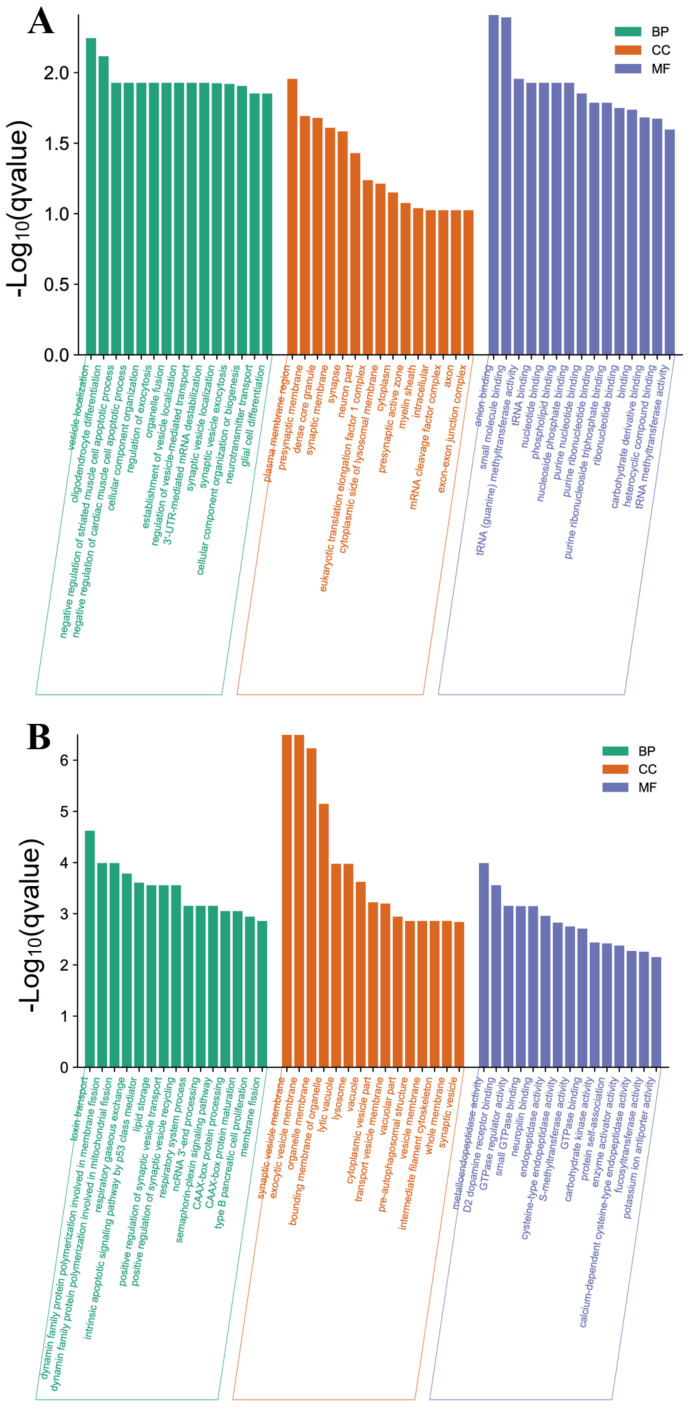
The top 15 enriched GO terms for the identified DEGs and the target genes of the DEMs in the LP-HF vs LP-LF. The *q*-value < 0.05 was used as a threshold to select significant GO terms. The X-axis and Y-axis represent the GO terms and the -log_10_ (*q* value) of the enriched genes, respectively. GO; Gene ontology, BP; biology process, CC; cellular component, MF; molecular function. (**A**) GO enrichment terms for the DEGs in the LP-HF vs LP-LF. (**B**) GO enrichment terms for the target genes of the DEMs in the LP-HF vs LP-LF.

**Figure 6 animals-12-03397-f006:**
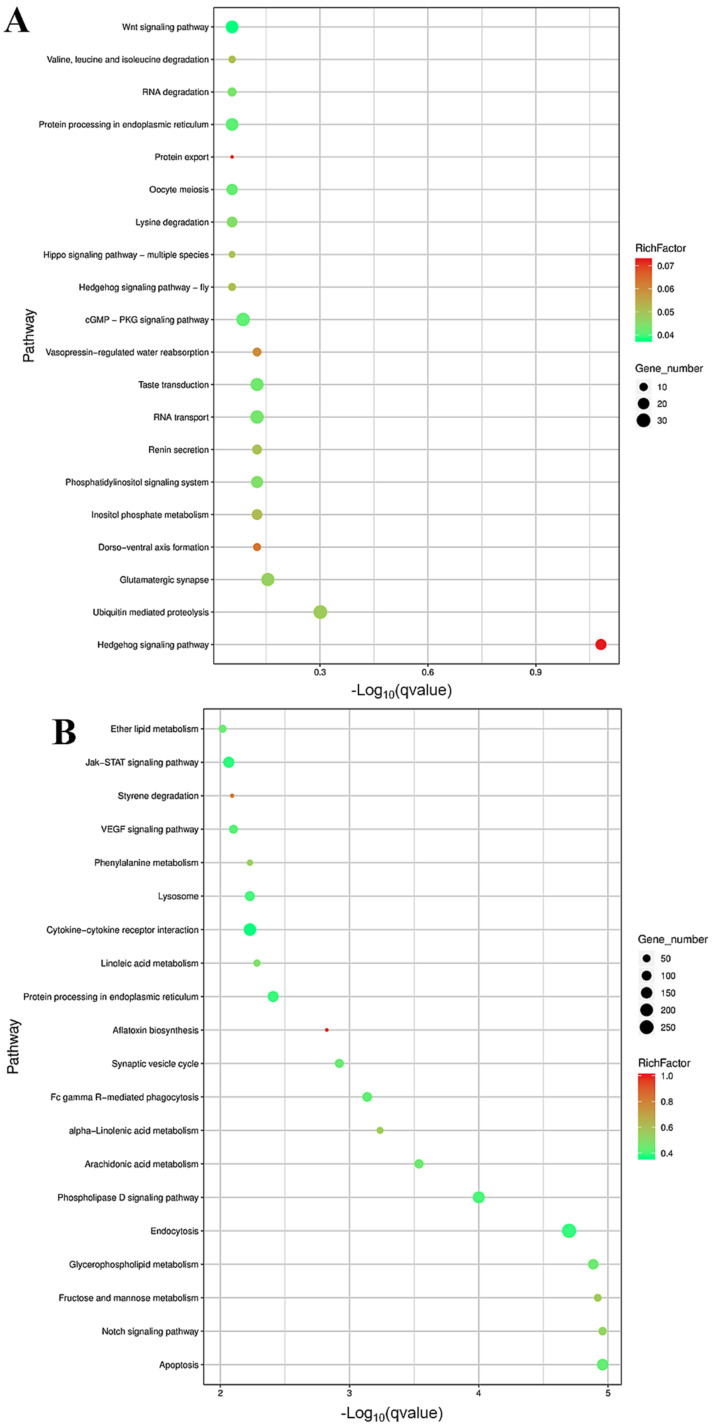
The top 20 enriched KEGG for the identified DEGs and target genes of the DEMs in the LP-HF vs LP-LF. The X-axis and Y-axis represent the -log_10_ (*q* value) of the enriched genes and the KEGG pathways, respectively. Rich factor refers to the ratio of the number of the differentially expressed gene and the number of the annotation genes enriched in this pathway term. The size of the circle in the figure indicates the number of the differential gene enrichment in the pathway. KEGG; Kyoto encyclopedia of genes and genomes. (**A**) KEGG enrichment pathways for the DEGs in the LP-HF vs LP-LF. (**B**) KEGG enrichment pathways for the target genes of the DEMs in the LP-HF vs LP-LF.

**Figure 7 animals-12-03397-f007:**
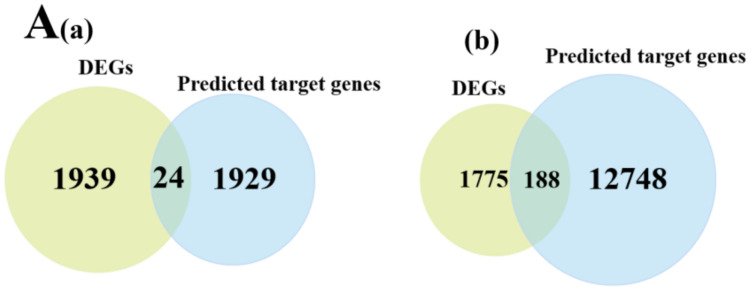
Overview of the mRNA-miRNA networks. (**A**) (**a**) Intersected genes in the LP-HF vs. LP-LF between the DEGs and the predicted target genes by the known miRNAs. (**b**) Intersected genes in the LP-HF vs. LP-LF between the DEGs and the novel miRNA-targeted genes. (**B**) Main hypothalamic interaction network in the LP-HF vs. LP-LF.

**Figure 8 animals-12-03397-f008:**
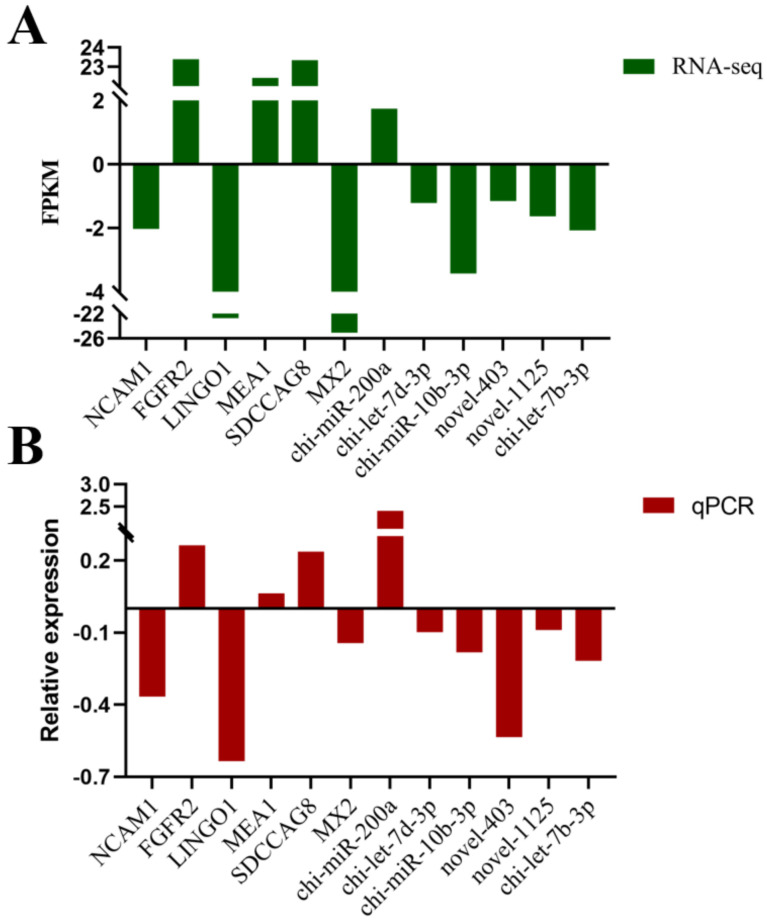
RT-qPCR validation of the mRNAs and miRNAs (**B**) identified by the RNA-seq(**A**) in the LP-HF vs LP-LF.

**Table 1 animals-12-03397-t001:** The primers of the mRNAs and miRNAs for the RT-qPCR.

Gene Name	Primer Sequences (5′-3′)	Accession No.	T_m_ (°C)
*NCAM1*	F: TGTGTGATGTGGTCAGCTCC	XM_018059775.1	60
	R: TGTGTGATGTGGTCAGCTCC		
*FGFR2*	F: ATGTCATCGTCGAGTACGCC	XM_018041428.1	60
	R: TAGGTGCACGACACCAAGTC		
*LINGO1*	F: CGCCTCAAGGTGTTGGAGAT	XM_018066520.1	60
	R: GACAGGGACGTCAGGTTGAG		
*MEA1*	F: AGGACGATGGCTGGCATAAG	XM_005696321.3	60
	R: CTTGGAGGGCCTTCTGTACC		
*SDCCAG8*	F: GGAGATTCTGGGGCAGTGTC	XM_013970102.2	60
	R: GTCACCTTCTCTCAGGGCAC		
*MX2*	F: CTCATTGACCTTCCCGGCAT	XM_018051644.1	60
	R: ATGGTCTCCTGCCTCTGGAT		
*RPL19*	F: ATCGCCAATGCCAACTC	XM_005693740.3	60
	R: CCTTTCGCTTACCTATACC		
chi-miR-200a	CAGTAACACTGTCTGGTAACG	-	60
chi-let-7d-3p	GCAGCTATACGACCTGCT	-	60
chi-miR-10b-3p	CGCAGACAGATTCGATTC	-	60
chi-let-7b-3p	AGCTATACAACCTACTGCCTT	-	60
novel-miR-403	GGGCCGGGCCT	-	60
novel-miR-1125	GCCCCTGGGCCT	-	60
U6 snRNA	CAAGGATGACACGCAAATTCG	-	60

**Table 2 animals-12-03397-t002:** Statistics for the cDNA library sequences of the hypothalamic tissue in the luteal phase.

Items	Clean Reads	Mapped Reads	Mapping Ratio	Q20(%)	Q30(%)	GC Content (%)
LP-HF1	98,815,860	95,807,657	96.96%	98.7;98.0	95.8;94.0	47.1;47.7
LP-HF2	102,074,738	99,172,836	97.16%	98.8;98.2	96.3;94.5	48.0;48.6
LP-HF3	106,891,624	102,778,831	96.15%	98.4;97.5	95.0;92.8	41.4;41.3
LP-HF4	107,291,396	103,863,460	96.81%	98.7;98.3	95.9;94.7	45.3;45.6
LP-HF5	118,968,066	115,301,498	96.92%	98.6;98.0	95.6;94.1	48.4;49.0
LP-LF1	124,269,432	120,367,153	96.86%	98.6;98.2	95.6;94.5	47.8;48.3
LP-LF2	104,606,848	101,179,431	96.72%	98.4;97.7	95.3;93.3	46.6;47.1
LP-LF3	106,329,382	102,901,653	96.78%	98.5;98.3	95.4;94.7	46.3;46.6
LP-LF4	124,634,642	120,894,187	97.00%	98.6;98.3	95.7;94.7	47.8;48.4
LP-LF5	117,361,866	113,601,480	96.80%	98.7;98.4	95.8;94.9	44.6;45.1

**Table 3 animals-12-03397-t003:** Statistics for the small RNA library sequences of the hypothalamic tissue from the luteal phase.

Items	Raw Reads	Clean Reads	Clean Ratio	Mapped Reads	Mapping Ratio	Q20	Q30	GC Content
LP-HF1	24,366,316	20,116,645	82.56%	19,590,837	97.39%	99.29%	97.69%	50.31%
LP-HF2	21,343,198	13,008,981	60.95%	12,775,878	98.21%	99.18%	97.55%	50.95%
LP-HF3	26,925,011	26,264,285	97.55%	25,677,239	97.76%	99.40%	97.89%	47.71%
LP-HF4	20,392,398	18,804,252	92.21%	18,460,000	98.17%	99.40%	97.87%	48.68%
LP-HF5	23,659,871	14,275,718	60.34%	13,988,821	97.99%	98.98%	97.19%	50.67%
LP-LF1	28,524,952	25,571,497	89.65%	24,371,807	95.31%	99.25%	97.27%	51.79%
LP-LF2	27,742,510	26,491,767	95.49%	25,784,762	97.33%	99.09%	96.69%	48.59%
LP-LF3	27,171,386	24,975,767	91.92%	24,300,180	97.30%	99.10%	96.60%	48.52%
LP-LF4	27,578,592	24,944,377	90.45%	24,450,755	98.02%	99.32%	97.67%	48.82%
LP-LF5	21,960,009	21,033,490	95.78%	20,562,709	97.76%	99.16%	97.40%	48.27%

**Table 4 animals-12-03397-t004:** GO terms related to the miRNA-mRNA co-expression network analysis.

GO ID	GO_Term	Gene Name	*q*-Value
GO:0043168	Anion binding	MX2, SEPT8, SNX17, RAB19, FES, RTEL1, ABCA3	0.0039
GO:0036094	Small molecule binding	MX2, SEPT8, IMPDH2, RAB19, FES, RTEL1, ABCA3	0.0041
GO:0098590	Plasma membrane region	ERC1, ADGRL1	0.0111
GO:0060627	Regulation of the vesicle-mediated transport	C1, TBC1D16	0.0118
GO:0000166	Nucleotide binding	MX2, SEPT8, IMPDH2, RAB19, FES, RTEL1, ABCA3	0.0118
GO:1901265	Nucleoside phosphate binding	MX2, SEPT8, IMPDH2, RAB19, FES, RTEL1, ABCA3	0.0118
GO:0016043	Cellular component organization	MX2, SUPT5H, SEMA4F, FLII, IMPDH2, ADGRL1, MRRF, MIEF2, C2, TBC1D16, FES, RTEL1, BCL9L, SDCCAG8	0.0118
GO:0005543	Phospholipid binding	SNX17, FES	0.0118
GO:0048284	Organelle fusion	TBC1D16	0.0118
GO:0071840	Cellular component organization or biogenesis	MX2, SUPT5H, SEMA4F, FLII, IMPDH2, ADGRL1, MRRF, MIEF2, C2, TBC1D16, FES, RTEL1, BCL9L, SDCCAG8	0.0124
GO:0017076	Purine nucleotide binding	MX2, SEPT8, RAB19, FES, RTEL1, ABCA3	0.0141
GO:0035639	Purine ribonucleoside triphosphate binding	MX2, SEPT8, RAB19, FES, RTEL1, ABCA3	0.0164
GO:0032555	Purine ribonucleotide binding	MX2, SEPT8, RAB19, FES, RTEL1, ABCA3	0.0164
GO:0051962	Positive regulation of the nervous system development	HEYL, ADGRL1	0.0164
GO:0016192	Vesicle-mediated transport	SNX17, ERC1, C2, TBC1D16	0.0178
GO:0032553	Ribonucleotide binding	MX2, SEPT8, RAB19, FES, RTEL1, ABCA3	0.0178
GO:0051179	Localization	MX2, FXYD5, SNX17, ERC1, MIEF2, C2, SLC29A3, PLIN2, TBC1D16, SRGAP3, FES, CLN3, ABCA3	0.0183
GO:0005488	Binding	MX2, SEPT8, SUPT5H, EGFL8, SNX17, FLII, HEYL, ERC1, IMPDH2, ADGRL1, MRRF, C2, TBC1D16, SRGAP3, RTEL1, TEAD2, ABCA3	0.0183
GO:0042734	Presynaptic membrane	ERC1, ADGRL1	0.0203
GO:0097367	Carbohydrate derivative binding	MX2, SEPT8, RAB19, FES, RTEL1, ABCA3	0.0208
GO:1901363	Heterocyclic compound binding	MX2, SEPT8, SUPT5H, HEYL, IMPDH2, RAB19, FES, RTEL1, TEAD2, ABCA3	0.0212
GO:0006810	Transport	MX2, FXYD5, SNX17, ERC1, MIEF2, C2, SLC29A3, PLIN2, TBC1D16, CLN3, ABCA3	0.0213
GO:0016050	Vesicle organization	TBC1D16	0.0213
GO:0045666	Positive regulation of the neuron differentiation	HEYL	0.0231
GO:0046907	Intracellular transport	MX2, SNX17, ERC1, MIEF2, TBC1D16	0.0231
GO:0097060	Synaptic membrane	ERC1, ADGRL1	0.0246
GO:0045202	Synapse	ERC1, ADGRL1	0.0261
GO:0051234	Establishment of the localization	MX2, FXYD5, SNX17, ERC1, MIEF2, C2, SLC29A3, PLIN2, TBC1D16, CLN3, ABCA3	0.0267
GO:0044087	Regulation of the cellular component biogenesis	MIEF2, FES	0.0296
GO:0051649	Establishment of the localization in the cell	MX2, SNX17, ERC1, MIEF2, TBC1D16	0.0304
GO:0097159	Organic cyclic compound binding	MX2, SEPT8, SUPT5H, HEYL, IMPDH2, RAB19, FES, RTEL1, TEAD2, ABCA3	0.0311
GO:0051960	Regulation of the nervous system development	HEYL, ADGRL1	0.032368404
GO:0035091	Phosphatidylinositol binding	SNX17, FES	0.0351
GO:0044801	Single-organism membrane fusion	TBC1D16	0.0371
GO:0097458	Neuron part	ERC1, ADGRL1, S100A1, MAPT	0.0373
GO:0043167	Ion binding	MX2, SEPT8, EGFL8, SNX17, IMPDH2, C2, RAB19, FES, RTEL1, MAN2A2, ABCA3	0.0429
GO:0051641	Cellular localization	MX2, SNX17, ERC1, MIEF2, TBC1D16, CLN3	0.0457

**Table 5 animals-12-03397-t005:** The KEGG pathways enriched in the miRNA-mRNA co-expression network analysis.

Gene Name	KEGG Pathway	Pathway ID
EVC	Hedgehog signaling pathway	ko04340
EDEM3	Protein processing in endoplasmic reticulum	ko04141
SEMA4F	Axon guidance	ko04360
DHDDS	Terpenoid backbone biosynthesis	ko00900
EPS15L1	Endocytosis	ko04144
GP1BB	Platelet activation; Hematopoietic cell lineage; ECM-receptor interaction	ko04611; ko04640; ko04512
ERC1	NF-kappa B signaling pathway	ko04064
IMPDH2	Purine metabolism; Drug metabolism—other enzymes	ko00230; ko00983
ERC1	NF-kappa B signaling pathway	ko04064
FLNC	MAPK signaling pathway; Focal adhesion	ko04010; ko04510
ABCC5	ABC transporters	ko02010
C2	Complement and coagulation cascades	ko04610
PLIN2	PPAR signaling pathway	ko03320
ACSS2	Carbon metabolism; Carbon fixation pathways in prokaryotes; Methane metabolism; Propanoate metabolism; Pyruvate metabolism; Glycolysis/Gluconeogenesis	ko01200; ko00720; ko00680; ko00640; ko00620; ko00010
VAV2	B cell receptor signaling pathway; Fc gamma R-mediated phagocytosis; Regulation of the actin cytoskeleton; Focal adhesion; cAMP signaling pathway; Chemokine signaling pathway; Fc epsilon RI signaling pathway; Leukocyte transendothelial migration; Natural killer cell mediated cytotoxicity; T cell receptor signaling pathway	ko04662; ko04666; ko04810; ko04510; ko04024; ko04062; ko04664; ko04670; ko04650; ko04660
PLCB2	Glutamatergic synapse; Phosphatidylinositol signaling system; Inositol phosphate metabolism; Renin secretion; cGMP—PKG signaling pathway; Wnt signaling pathway; Dopaminergic synapse; Long-term potentiation; Gap junction; Apelin signaling pathway; Calcium signaling pathway; Phospholipase D signaling pathway; Rap1 signaling pathway; Sphingolipid signaling pathway; Adrenergic signaling in cardiomyocytes; Vascular smooth muscle contraction; Gastric acid secretion; Pancreatic secretion; Salivary secretion; Aldosterone synthesis and secretion; Estrogen signaling pathway; Glucagon signaling pathway; GnRH signaling pathway; Insulin secretion; Melanogenesis; Oxytocin signaling pathway; Thyroid hormone signaling pathway; Thyroid hormone synthesis; Circadian entrainment; Endocrine and other factor-regulated calcium reabsorption; Chemokine signaling pathway; NOD-like receptor signaling pathway; Platelet activation; Cholinergic synapse; Long-term depression; Retrograde endocannabinoid signaling; Serotonergic synapse; Inflammatory mediator regulation of TRP channels; Phototransduction-fly	ko04724; ko04070; ko00562; ko04924; ko04022; ko04310; ko04728; ko04720; ko04540; ko04371; ko04020; ko04072; ko04015; ko04071; ko04261; ko04270; ko04971; ko04972; ko04970; ko04925; ko04915; ko04922; ko04912; ko04911; ko04916; ko04921; ko04919; ko04918; ko04713; ko04961; ko04062; ko04621; ko04611; ko04725; ko04730; ko04723; ko04726; ko04750; ko04745
SRGAP3	Axon guidance	ko04360
FES	Axon guidance	ko04360
EPS15L1	Endocytosis	ko04144
COL18A1	Protein digestion and absorption	ko04974
MAPK15	IL-17 signaling pathway	ko04657
COL11A2	Protein digestion and absorption	ko04974
CLN3	Lysosome	ko04142
COL13A1	Protein digestion and absorption	ko04974
TXNRD3	Selenocompound metabolism	ko00450
MAPK8IP3	MAPK signaling pathway	ko04010
MAN2A2	N-Glycan biosynthesis; Various types of N-glycan biosynthesis	ko00510; ko00513
MAPT	MAPK signaling pathway	ko04010
SYVN1	Ubiquitin mediated proteolysis; Protein processing in the endoplasmic reticulum	ko04120; ko04141
TEAD2	Hippo signaling pathway-multiple species; Hippo signaling pathway-fly; Hippo signaling pathway; MAPK signaling pathway-yeast	ko04392; ko04391; ko04390; ko04011
MAPT	MAPK signaling pathway	ko04010
L1CAM	Cell adhesion molecules (CAMs); Axon guidance	ko04514; ko04360
ABCA3	ABC transporters	ko02010

## Data Availability

The original contributions presented in the study are included in the article and Appendix A, further inquiries can be directed to the corresponding authors.

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
