# Peer review of "Integrated Transcriptome Analysis Reveals the Crucial mRNAs and miRNAs Related to Fecundity in the Hypothalamus of Yunshang Black Goats during the Luteal Phase"

_animals, 2022, doi:10.3390/ani12233397_

Round 1
Reviewer 1 Report
The paper is beyond my knowledge when it comes to lab technology, software used and physiology. I understand however that what has been done is of interest for changing prolificacy of goats (and probably many other mammals).
Whether this should lead to selection for specific alleles is not so clearly treated, and may be too early to say yet.
English is mostly clear and good, but some improvements are required. A few examples are given below.
Alterations in hypothalamus function, which also 50
affects reproductive activity, including follicular development, ovulation, and oviposition 51
->
Alterations in hypothalamus function, also 50
affects reproductive activity, including follicular development, ovulation, and oviposition 51
Since including the Booroola fe- 58
cundity gene (FecB) was identified in Booroola sheep [8], a series of studies had been car- 59
ried out on the major genes controlling sheep and goat follicle development, estrus and 60
ovulation by using molecular biology techniques
->
Since the Booroola fe- 58
cundity gene (FecB) was identified in Booroola sheep [8], a series of studies have been car- 59
ried out on the major genes controlling sheep and goat follicle development, estrus and 60
ovulation by using molecular biology techniques
had -> has 61
were -> are 64
... and possibly more similar ....
agouti ?-> Agouti 72 and similar for other names ?
Kispeptin -> Kisspeptin 77 + 78
Author Response
Dear Reviewers:
First of all, thank you very much for your suggestions on the revision of my manuscript
My replies to the comments on Reviewer 1 are as follows:
- For some improvements of English sentences in the manuscript, including several examples given by the teacher, I have modified them in the manuscript.
- agouti? - > Agouti and similar for other names?
The question about the correct name of agouti-related peptide, you can refer to literature Gouveia A, de Oliveira Beleza R, Steculorum SM. AgRP neuronal activity across feeding-related behaviours. Eur J Neurosci. 2021 Nov;54(10):7458-7475. doi: 10.1111/ejn.15498. A detailed explanation of the AgRP can be found on the front page of the article.
Thanks again for your suggestions on the revision of my manuscript.

Reviewer 2 Report
In this manuscript,RNA-sequencing was used to analyze the mRNA and miRNA expression profiles of hypothalamic tissues between high-fecundity and low-fecundity goats in the luteal phase,lots of DEGs were found in RNA-seq and miRNA-seq,The results could provide a reference for molecular regulation mechanism of goats high-fecundity. I would suggest that some questions and suggestions need to be addressed in this research before its publication.
1. Average litter size of experimental goats were given in Materials. However, the details of reproduction phenotype was not mentioned, size of every litter, genetic background of experimental animals could be supplied.
2. How is the results of cluster dendrogram of the samples.
3. In the Introduction, the last sentence of paragraph 2(line 80-82) could be transfer to the beginning of paragraph 3 according to the theme of each paragraph.
4. As a conclusion, several DEGs, such as 517 CYP19A1, NCAM1 and FGFRs and miRNA-mRNA pairs (including MEA1 was co-ex-518 pressed with novel-972, novel-125 and novel-403) may play important roles in the reproductive regulation of goats. However, there was no known genes and miRNA such as FecB, RBP4, miR-30b etc., was screened for analysis of differential expression and functional enrichment, how does the author explain the results.
Author Response
Dear Reviewers:
First of all, thank you very much for your suggestions on the revision of my manuscript
My replies to the comments on Reviewer are as follows:
- Average litter size of experimental goats were given in Materials. However, the details of reproduction phenotype were not mentioned, size of every litter, genetic background of experimental animals could be supplied?
We randomly selected goats from the same experimental base, 5 ewes with high-fecundity and 5 ewes with low-fecundity goats were selected for full transcription sequencing. The main purpose was to screen differentially expressed genes in hypothalamus tissues of goats with different reproductive performance. Statistics were not made on size of every litter, but only on the number of lambs.
- How is the results of cluster dendrogram of the samples.
I made the cluster dendrogram of the samples according to the average litter size number of the ten samples of this study, as shown in the figure below. It can be seen that the samples information of high yield group and low yield group are obviously different. But the figure can't explain something in the text, so I will show it to the reviewer here.
- In the Introduction, the last sentence of paragraph 2(line 80-82) could be transfer to the beginning of paragraph 3 according to the theme of each paragraph?
It has been modified as required.
- As a conclusion, several DEGs, such as CYP19A1, NCAM1 and FGFRs and miRNA-mRNA pairs (including MEA1 was co-expressed with novel-972, novel-125 and novel-403) may play important roles in the reproductive regulation of goats. However, there was no known genes and miRNA such as FecB, RBP4, miR-30b etc., was screened for analysis of differential expression and functional enrichment, how does the author explain the results.
The differentially expressed miRNAs correspond to differentially expressed genes in the miRNA and mRNA co-expression network, so the range of known genes and miRNAs after screening is smaller. The known genes related to reproduction, such as RBP4, are also differential expression with different reproductive performance in my data. However, there was no corresponding differentially expressed miRNA, so the gene was not screened out for my network interaction analysis. In addition, the FecB gene is currently found in the reproductive performance of sheep. My experimental animal is goat, and the experiments on the hypothalamus tissue of goat as the research object have hardly been found at present. Therefore, due to the differences in species and tissues, the genes screened will be somewhat different from the classical known genes related to reproduction.
Thanks again for your suggestions on the revision of my manuscript.

Reviewer 3 Report
Dear Author,
animals-2011841 Integrated transcriptome analysis reveals the crucial mRNAs and miRNAs related to fecundity in the hypothalamus of Yunshang black goats at the luteal phase. The manuscript is well written, however, I`m not a good judge in this issue because I`m not an English native speaker. The manuscript touches on important things – reproduction in goats and shows new information about the transcriptome of the hypothalamus during the luteal phase in low- and high-fecundity goats. I have one main concern because goat breeding is not exactly my area of expertise and maybe I do not exactly understand, but why did authors use these experiments on goats between 3-5 years? Why weren`t they the same age? Probably 5 years goats had more parity than three-old. Nevertheless, if authors have the confirmation that age and parity did not affect hypothalamus transcriptome and expression profiles of microRNAs, they should attach here this information. Moreover, authors should indicate which was the average age per low- and high-fecundity group, besides I have a few minor comments:
Abstract
LLine 24 it is not exactly clear what does mean novel-972, novel-125 and novel-403 – because in the abstract section is just a little place to describe, maybe change this for three novel miRNA
Introduction line 68 - before the sentence “miRNA (miRNA) can bind to complementary mRNA”, should insert one sentence something like that ”on the other hand different molecules playing an epigenetic role such as miRNA were also investigated in this context”
Material and methods
If goat age and number of parity have no effect on hypothalamus transcriptome and miRNA profile please insert some confirmation. Because 5-years goats had probably more pregnancies than three year old. It should be dissolved this problem.
Line 134 - Which RIN value was accepted?
Line 181 – which reference was used in this functional analysis for transcriptome and miRNA? Human, pig?
Results are presented clearly. However, sides 14 and 18 are the same in Figure no 7.
Discussion
Based on miRNA-mRNA network analysis. Here should be indicated gene hubs, which are the most important based on this study. Which should be underlined also in the conclusion section.
Author Response
Dear Reviewers:
First of all, thank you very much for your suggestions on the revision of my manuscript
My replies to the comments on Reviewer are as follows:
a. I have one main concern because goat breeding is not exactly my area of expertise and maybe I do not exactly understand, but why did authors use these experiments on goats between 3-5 years? Why weren`t they the same age?
The subjects in the manuscript were 3-5-year-old meridians, all of whom gave birth to two litters, and all of whom were actually around four years old, not the extreme three- and five-year-olds. Their average age is four. According to the teacher's advice, I have added this information into the manuscript.
b. For some improvements of sentences in the manuscript, I have modified them in the manuscript.
Thanks again for your suggestions on the revision of my manuscript.
We deeply appreciate your consideration of our manuscript, and we look forward to receiving comments from the reviewers. Correspondence should be directed to Mingxing Chu at following address.
c. Which RIN value was accepted?
They're all acceptable. We just tested the concentration, purity and degradation of the samples with different instruments.
Thanks again for your suggestions on the revision of my manuscript.

Round 2
Reviewer 2 Report
nothing
Reviewer 3 Report
The authors addressed almost all my suggestions so I think that the manuscript can be published in the present form.